# A Driver Gaze Estimation Method Based on Deep Learning

**DOI:** 10.3390/s22103959

**Published:** 2022-05-23

**Authors:** Sayyed Mudassar Shah, Zhaoyun Sun, Khalid Zaman, Altaf Hussain, Muhammad Shoaib, Lili Pei

**Affiliations:** 1Information Engineering School, Chang’an University, Xi’an 710061, China; mudassarshah@chd.edu.cn (S.M.S.); khalidzaman@chd.edu.cn (K.Z.); peilili@chd.edu.cn (L.P.); 2Department of Computer Science and IT, The University of Agriculture Peshawar, Peshawar 25000, Pakistan; altafscholar@aup.edu.pk; 3Department of Computer Science and IT, CECOS University, Peshawar 25000, Pakistan; mshoaib@cecos.edu.pk

**Keywords:** advanced driver-assistance technique, You Only Look Once (YOLO), Inception-v3, CNN, InceptionResNet-v2

## Abstract

Car crashes are among the top ten leading causes of death; they could mainly be attributed to distracted drivers. An advanced driver-assistance technique (ADAT) is a procedure that can notify the driver about a dangerous scenario, reduce traffic crashes, and improve road safety. The main contribution of this work involved utilizing the driver’s attention to build an efficient ADAT. To obtain this “attention value”, the gaze tracking method is proposed. The gaze direction of the driver is critical toward understanding/discerning fatal distractions, pertaining to when it is obligatory to notify the driver about the risks on the road. A real-time gaze tracking system is proposed in this paper for the development of an ADAT that obtains and communicates the gaze information of the driver. The developed ADAT system detects various head poses of the driver and estimates eye gaze directions, which play important roles in assisting the driver and avoiding any unwanted circumstances. The first (and more significant) task in this research work involved the development of a benchmark image dataset consisting of head poses and horizontal and vertical direction gazes of the driver’s eyes. To detect the driver’s face accurately and efficiently, the You Only Look Once (YOLO-V4) face detector was used by modifying it with the Inception-v3 CNN model for robust feature learning and improved face detection. Finally, transfer learning in the InceptionResNet-v2 CNN model was performed, where the CNN was used as a classification model for head pose detection and eye gaze angle estimation; a regression layer to the InceptionResNet-v2 CNN was added instead of SoftMax and the classification output layer. The proposed model detects and estimates head pose directions and eye directions with higher accuracy. The average accuracy achieved by the head pose detection system was 91%; the model achieved a RMSE of 2.68 for vertical and 3.61 for horizontal eye gaze estimations.

## 1. Introduction

According to the World Health Organization (WHO), automobile crashes are one of the world’s leading causes of death, with road injuries ranking among the top ten leading causes of death worldwide [1]. In recent years, globalization has led to an increase in the number of people worldwide who have died in these types of car crashes, which is a cause for concern. According to [2], driver distractions are responsible for approximately 81% of all automobile crashes that occur on public highways. According to the study, more than 1.98 million people die from traffic-related injuries each year in the United States alone. Every year, hundreds of thousands of people are killed or seriously injured in car crashes, resulting in significant human and financial losses [3]. Researchers have discovered that motorists who are capable of signaling drivers and are traveling with one or more passengers are less likely (29–41%) than those traveling alone to cause collision-related damage. This research resulted in the development of an ADAT [4], which has gained widespread acceptance in both the automotive industry and academia due to its effectiveness. These systems are designed to aid drivers in making decisions while driving to improve overall vehicle and road safety. The ADAT is becoming increasingly common in mid-range and lower-end vehicles [5], bucking the current trend. Many studies are being conducted to advance the development of systems based on analyses of the physical characteristics of drivers in vehicle crashes (to improve the safety of the drivers involved).

In several recent studies, researchers discovered that driver fatigue could impact driving performance (comparable to that of alcohol in some circumstances). Driver fatigue monitoring systems have been researched and developed for more than two decades and they are now being used worldwide. An investigation into the use of a single-channel electroencephalographic device to monitor driver fatigue was recently conducted [6], with particular attention being paid to the saccadic movement of the eyes. This is demonstrated in [7], where the driver’s blinks were analyzed to determine his or her level of fatigue, using a standard USB camera as the input device. In other research, there was precedence in using velocity as a reference index of fatigue; this is an example of such a technique being used in other research. Implementing an automatic system to warn or inform the driver of an impending crash must occur after an important design flaw is identified and addressed before moving forward with implementing the system itself. A recent investigation discovered that sloppy engineering in warning systems could hurt the driver’s ability to operate the vehicle safely. Regarding drivers, it has been demonstrated that increasing their workloads while driving causes them to lose sight of their surroundings, which is dangerous. As previously mentioned, the accumulation of unnecessary information can be overwhelming, leading to apathy and, eventually, deactivation of the warning system. This project proposes a technique using computer vision and deep learning technologies to detect various types of head movements and eye gaze directions via a simple RGB camera. There is a hierarchy of alarms [6,8]; as a result, the system will only alert the driver when necessary. An ADAT can only warn a driver in two situations: when the driver is not paying attention to the road and when the situation is extremely hazardous.

In these situations, if the driver is not paying attention while driving, the ADAT will not issue any warnings to the driver. Following the application of this development method, we successfully developed an entirely new proposal that was in sync with the most recent technological advancements. According to scientists, a camera-based eye tracker is used in the proposed system and the data collected could be used for a variety of purposes, depending on the circumstances. Drowsiness and fatigue, among other things, could be detected using this technology. A modern ADAT provides drivers with assistance to improve overall safety in and around their vehicles and on the highway. The Prometheus Project (Program for European Traffic of Maximum Efficiency and Unprecedented Safety), which began in 1986 and was tasked with developing solutions to traffic congestion, brought together several European automotive companies and research institutes to collaborate on the project. A watershed moment in the history of the automotive industry was marked by this collaboration, which was widely hailed as a success. To begin the research, it was initially necessary to wait until the last two decades, when significant progress in this type of research has been made. This was necessary because the technology was still in its infancy throughout those years, making it difficult to begin the research. In the last few years, scientists have concentrated their efforts on this area, achieving great success in improving road safety. 

The development of an ADAT is currently underway to increase road traffic safety. Its capabilities include the ability to intervene in a variety of ways to avoid potentially hazardous situations [6], and to improve driver and vehicle safety [9] to accomplish these objectives. Considering that human error is responsible for 93% of all traffic crashes [10], it is essential to research technological areas related to the development of an increasingly powerful and comprehensive ADAT. When it comes to driver-centered assisted driving technologies [11], the information gathered from the motorist is used to provide the driver with assistance. If a driver information system does not incorporate driver information into its design, it will be impossible to detect vehicle departures that are not intentional, such as those detected by lane departure detection techniques. The technique may be unable to distinguish between intentional and unintentional vehicle departures during a crash, resulting in a potentially fatal crash. We can infer that the vehicle’s departure is not planned if the driver appears to be distracted or sleepy during the incident. If the investigating officer determines that it is necessary, an official written warning can be issued to the driver due to the incident. This paper proposes a real-time (non-intrusive and non-invasive) gaze tracking technique for drivers. 

This technique continuously determines where the driver is looking, providing the information required (e.g., the driver’s concentration level) to the advanced driver-assistance system. The system can then use this information to assist the driver in various ways. Installing a gaze tracking system in a vehicle is required for an ADAT and other in-vehicle warning systems to determine whether or not the driver needs to be informed about a specific risk that may arise during the journey. Using a visual area analysis to determine if the driver is distracted is the underlying concept; warning systems on the vehicle will be activated due to this determination. To avoid the need for time-consuming calibration procedures, users can be instructed to look at specific points on the screen to adjust the intrinsic parameters of the eye model. Additionally, integrating a gaze tracking system model into a driver-centered active safety system that is data-fusion based, in which the driver’s gaze is tracked in real-time, is being explored. 

One of the most common causes of car crashes is a driver’s lack of concentration. These crashes could be avoided if alert and monitoring systems are in place. Developing an intelligent monitoring system to detect driver inattention based on eye gaze is proposed. The driver’s gaze also provides sufficient information to predict whether or not the driver is looking at the road. It captures facial images with the help of a camera. Computer vision and deep learning technologies enable us to (1) detect if the driver is in an inactive state and (2) pay attention to the driver. The proposed system is about developing a system that will predict the head pose and eye gaze information deployed in a car or truck. The predicted information of the driver will be input into an automatic driver-assistance system. In case of any abnormal event, or if the driver is detected, an automatic vehicle control system will take control of the vehicle and reduce its speed to park the vehicle, avoid the road, and provide safety to human lives and public property.

The key objectives/contributions of the proposed study are as follows;

We propose a novel deep learning approach to detect and estimate the driver’s gaze.We propose the CNN regression model, elaborated with other deep learning models, to detect and extract the features from the given dataset and the detection of real-time data.We propose an ADAT technique with the deep learning InceptionResNet-v2 to observe and detect the data from input and output processes.We evaluated the proposed work based on MSE, RAE, and RMSE performance evaluation metrics to check for accuracy and loss.

The remainder of the paper is organized as follows: a high-level abstraction is used to provide an overview of the development of driver inattention monitoring techniques, as well as related work for intelligent vehicles, such as gaze tracking techniques. Section 2 consists of a detailed literature review. Section 3 is the research methodology section, which contains dataset descriptions and frameworks of the proposed models along with the mechanisms of model training and hyperparameter tuning for optimal feature weights. Section 4 contains information about the experimentation performed, performance evaluation, model performance comparison, and results. Section 5 concludes the paper. 

## 2. Literature Review

Currently, available research focuses on which gaze direction of the driver could be divided into single and multiple cameras. Because of the maximized motion (e.g., rotation of the head) of the driver, detecting the driver’s gaze with a single camera is more difficult in a vehicle environment [10,11,12,13,14,15,16,17,18,19,20,21]. As the driver rotates his or her head in numerous directions or gazes at an object with only his/her eyes moving while keeping his/her head stable, the accuracy of a single-camera method tends to deteriorate. As a result, researchers have looked into methods for addressing these flaws involving multiple cameras. Dual cameras are frequently mounted near the vehicle’s A-pillar or dashboard when they are used. In a previous study [11], cameras with dual lenses were used to detect the gaze of the driver and to estimate his/her inattention. However, when using the available Smart Eye Pro 4.0 eye tracker at the commercial level, the accuracy of real-time eye detection performance was different, depending on the price of the equipment. Additionally, there were times when user calibration for glasses wearers was not supported [12]. Furthermore, while the fatigue and inattention algorithms directly depended on the driver’s accuracy (of the gaze detection), only 23% of the dataset (for the overall distance throughout the research) showed a low detection of gaze reliability. In a previously published study, SVM was used to detect driver inattention in real-time [22]. However, each driver must undergo initial calibration for eye tracking, which takes about 5 to 15 min. Additionally, because wearing glasses or eye makeup can affect tracking accuracy, drivers are not permitted to do so. The authors of [3] used a three-lens camera to track the driver’s facial features in order to compute the driver’s head position. However, the facial feature tracking accuracy and recognition of the position were determining factors in the performance.

The three-lens camera did not require any additional adjustment compared to stereo cameras. On the other hand, the proposed framework was used independently, and the results from the three-lens camera necessitated further analysis. Furthermore, it was difficult to detect the changes in eye poses caused by pupil movements since it only measured the facial movements of the driver and ignored the pupil movements. The authors of [23] proposed a technique to capture the facial images of the driver via two USB cameras. On the other hand, a camera with two stereos required to be adjusted to a map of depth. Furthermore, their research was conducted in an office—an indoor scenario setting—instead of a vehicle. The gaze zone was estimated using the driver’s head pose in a previous study [24]. Since the live photo was taken from the drive with the help of a visible-light camera without the illumination of NIR, the driver’s performance was affected by variations in illumination. Furthermore, while the driver’s gaze accuracy was high, the gaze region of the front was segregated into approximately eight regions, which resulted in multiple regions that were excluded from the measurement of the accuracy. Furthermore, the gaze was tracked using the driver’s head movements rather than the position of the pupil center; accuracy of the gaze-tracking suffered at the time—only the pupils from the eye movements and the head did not (they remained stationary). The authors of [25] proposed the gaze zone of the driver based on the association among the head directions of the driver, the iris, and the eye. The framework for the continuous movements of head estimations projected in [26] was utilized to achieve an appropriate estimation of the head pose selectively with the utilization of many cameras. The methods used to track the driver’s gaze, via multiple cameras, are extremely accurate. However, as the input images grow in proportion to the cameras, the required processing time will grow. As a result, the current study proposes using a single deep residual network to combine two images acquired from two cameras, with NIR and with the illuminator, into single three-channel images, tracking the eyes of the driver in the vehicle (ResNet).

From the above, it appears that more investigations are required into eye gaze estimations with numerous approaches and multiple angles while a driver sits in the vehicle. Due to the various measurement methods used, no general conclusions can be drawn. Eye contact-seeking can be inferred from capacities such as the orientation of the head, which can result in reports, by examining standing on the side of the road, or could be captured by cameras inside or outside the vehicle [7,8,13,27,28]. The authors of [10] proposed an approach that involved walkway pedestrians looking at a vehicle’s arriving direction for one second; in this way, they might experience eye contact. This was based on a video taken in an urban driving scenario. On the other hand, the orientation of the head does not determine (as compared to the rotation of the eye). In their paper, they proposed an eye-tracking system to detect eye contact. This system could determine the location and direction of the gaze without clearly ‘requesting’ road users or relying on third-party annotations. Numerous research studies have been conducted on this approach, concerning a driver’s eye gaze without head orientation or looking at the pedestrian [25,26]. Drivers, for instance, look at a cyclist’s face first; after that, they look at the other parts of the body, according to Walker [14] and Walker and Brosnan [15]. The authors of [16] proposed an eye-tracking system to examine the gaze estimation of a driver and found that in the majority of interactions, pedestrian body postures/movements and eye gazes were enough to decide passage engagements without the use of eye contact or hand gestures. The authors developed a novel method [17], in a scenario that used simulations and eye-tracking, and found that the pedal response of a driver (to turn the brakes on) was categorized by the eye gaze estimation at a level of 0.4–2.4 s regarding the earlier pedestrian. The authors of [29] tackled a system named eye-tracking in which contributors watched real-time videos of traffic from the driver’s perspective, pressing a button whenever they saw hazardous activity. According to the writers, the drivers fixated on walkers on the road more frequently compared to those on ‘restrictions’. The authors of [30], in another research study conducted on the eye movements of pedestrians, proposed a study in which the eye-tracking movements of pedestrians were utilized in a parking lot during interactions with vehicles; they discovered that walkers often pursued and made eye contact with the drivers. The authors of [18] used a handheld slider to track walkers at a restriction; they measured the willingness of pedestrians regarding incoming vehicles. Even though approaching car interiors are shady, and replications on the windshields make eye contact problematic, pedestrians still looked at the windshields for evidence of the intentions of the drivers when there were vehicles close by. The authors of [19] proposed a study for pedestrians in which eye contact and tracking were discovered. They noticed a flaw in this approach because of the physical movements and the eye gaze movements whenever a pedestrian came in the way or near the road. This limitation could be overcome by combining image recognition with eye-tracking. An important additional warning in the study by [19], e.g., [19,20,29], involving eye tracking in traffic, is that merely one viewpoint (the driver’s or the pedestrian) was dignified, providing a non-complete since eye contact is a shared occurrence [31,32]; a dual eye contact approach for study (in social kinds of interactions) was proposed. Detecting the gaze of only one of the two get-togethers provides information regarding the get-together’s desire for eye contact; nevertheless, this does not indicate whether or not eye contact has been recognized. The authors of [33] reported similar issues with operational types of investigation for eye tracking in their literature reviewing human–human interactions. The approaches that are used to detect eye gaze and pedestrian eye contact could fill this gap in the literature.

## 3. Methodology

As depicted in Figure 1 below, the overall framework of our research is laid out. The development of the human gaze dataset involved the participation of 30 subjects in the dataset development process. By looking in different directions, a total of 7 gazes were selected for analysis. The development of a face detector technique occurred in the second phase. The proposed detector is proposed to be specifically designed for a car—an indoor face detection with efficiency and accuracy; the developed face detection technique is extremely robust and performs well in extreme conditions, such as low lighting and face obstacles. In this research work, the most recent and highly accurate convolutional neural network model is trained to detect the driver’s gaze in an indoor car scene, which is a challenging task.

### 3.1. Custom Dataset Creation

A high-resolution camera was used to create a benchmark dataset for gaze detection, which will be used in future research. Participation in the dataset creation process was limited to 30 individuals. In this study, all of the subjects (drivers) were male and between 25 and 40. They were bearded or had no beard; wore glasses or not; or wore caps or not. A professional photographer photographed, shot, and filmed the subjects during the indoor car scenes. Face identification and gaze detection techniques could be rendered inaccurate by various factors, including physical obstructions and significant changes in conditions. Two separate datasets were developed, i.e., (1) a gaze detection dataset that was composed of seven classes of drivers’ head poses and (2) an eye gaze direction containing the images of eyes looking at various vertical and horizontal angles. The gaze detection dataset consists of seven classes; the details of the data are described in Table 1. The second dataset consists of images used for eye gaze estimation; using a single image, the horizontal and vertical gazes were estimated. The total number of eye gaze angles for each plan was nine; this doubled for horizontal and vertical plans. The gaze estimation dataset details can be seen in Table 2. Training and validation of the proposed models took place on a local dataset that was created by the researchers, as well as training and validation on benchmark driver emotion recognition datasets from other sources. The Figure 2 is regarding the face direction point of the dataset. Similarly, the Figure 3 is also regarding eye gaze directional point. And Figure 4 is regarding the random gazes recorded for the developing benchmark dataset.

### 3.2. Face Detection Using YOLO

We propose a method called YOLO-face, which is intended to detect faces with varying scales by combining several techniques. We anticipate that YOLO-face will have a detection speed that is comparable to YOLO in terms of detection speed. The YOLO-V4 framework serves as the foundation for the YOLO architecture. To create the model (extra proper) for multi-scale face recognition applications, we improved the existing YOLO-V4 by replacing the feature learning CNN model from darknet to Inception-v3 to extract robust features and improve the face detection accuracy.

#### 3.2.1. YOLO-V4 Network

The YOLO-V4 network, a single-stage object detector–object detection algorithm, was applied. Several object detection techniques–feature extraction networks are critical; the method’s detection accuracy is highly dependent on the feature abstraction system’s capacity to extract rich structures from the input dataset. If a network can extract more features, it will do high-precision detection. Inception-v3 CNN replaced CSP-Darknet-53 as a feature extraction system in YOLO-V4. The YOLO-V4 network delivers quick detection but struggles to recognize tiny objects, resulting in incorrect detection. As a result, additional enhancements are required to create a mask detection algorithm capable of recognizing tiny items, such as a mask on the face. The suggested work improves the feature extraction system detector, YOLO-V4, for unique mask detection jobs by integrating a longitudinal pyramid assembling component and an extra detector YOLO layer to identify disguises with excessive precision and accuracy.

This study utilized the complete-IoU (C-IoU) loss function presented in [14] to promote accurate regression and quicker convergence. The proposed network’s anchor frame is indomitable by a k-means++ grouping algorithm tailored to the mask detection dataset. The SPP element, the convolutional layer, and the third detection YOLO layer were added to YOLO-V4 to build the improved network YOLO-V4. The original modified YOLO-V4 feature extraction network had 21 convolutional layers and 3 maximum pooling layers. For detection, the improved YOLO-V4 employs two detection YOLO layers that are joined after the feature abstraction layer. We improved the modified YOLO-V4 feature extraction network to upsurge its total detection accurateness for disguise recognition. After the feature extraction layer, we included an SPP element to abstract rich features and a detector YOLO layer to allow the system to recognize tiny objects. 

The SPP element decreases computational convolution by avoiding the convolutional layer from handling structures that are removed repeatedly, enhancing the network’s speed. For YOLO-V4, the proposed feature extraction network has 25 layers of convolutional and 3 maximum layers of pooling. An SPP element, consisting of 3 maximum layers of pooling of sizes, 1 × 1, 3 × 3, and 5 × 5 × 2048, was also used after reducing the dimension of the feature map from 26 × 26 × 512 to 13 feature extraction layers. After the feature extraction network, 3 YOLO layers of detection were deployed for detection. The proposed system employed the YOLO detection layer to forecast the bounding box. For each item, the bounding boxes in a class in the mask detection dataset were also predicted using anchor frames. The bounding box, object score, anchor point, and class prediction were all handled by the network’s three YOLO detection layers. 

The size of the filter before every layer of YOLO was computed by multiplying (classes + 5) by 3, the number of classes in this work was 4, and the filter size was 27. Compared to the YOLO-V4 network, the work proposed here added four kinds of convolutional layers along with the SPP element, with three maximum layers of pooling, all of which aided in extracting rich information from the data input. In addition, the third detection YOLO layer used a high-dimensional map of features (size 52 × 52 × 27) to increase object localization and prediction accuracy. Figure 1 depicts the planned YOLO-V4-SPP network’s full design.

#### 3.2.2. Anchor Boxes Appropriate for Face Detection

The scales and ratios of anchor boxes are very important hyperparameters to consider when performing object detection. The shapes of the anchor boxes should be closely related to the targets that are identified. To provide the greatest number of possibilities for general object detection, anchor boxes should be designed so that they contain as many possibilities as possible. If you look at an image, most of the faces appear to have higher heights than their widths, which is intuitively correct. A consequence of this should be that the shapes of the anchor boxes used for face detection should differ from the shapes used for general object detection. We put together two different anchor boxes intending to determine which ones would be the most appropriate for facial recognition. At least one instance used YOLOv3 anchors that were converted from flat to slim boxes and were taken directly from the original YOLOv3. We use the term “flat” to describe the fact that the heights of the boxes were less than their widths, and slim boxes were both narrow and tall in proportion to their widths. To obtain the dimensions of bounding boxes for other types of anchors, such as those used in the following versions of YOLOv2 and YOLOv3, the k-means clustering on the WIDER FACE training dataset was performed on the training dataset as illustrated in Figure 5. 

The following is the procedure to be followed: it is necessary first to determine the number of seeds (k) that will be used, and then to select a random number of anchor boxes as initial clustering centers, with the IoUs of both the initial clustering centers and all other anchor boxes being calculated afterward. K classes are created by categorizing all face labels based on the IoU and the distance between anchor boxes. To determine the new cluster centers, calculate the k class anchor box size’s mean and divide the result by the number of classes in each class. The process should be repeated until convergence is achieved. In our experiments, we began with a total of nine cluster centers to test our hypotheses (k). Because of this, the horizontal anchor boxes were flipped vertically to allow for face detection. The following: (156, 198), (3, 3), (30, 61), (45, 62), (59, 119), (90, 116), and (156, 198) are among the final nine anchor shapes.

#### 3.2.3. Evaluating Face Detection Models

Mean average precision (mAP) was used to evaluate the proposed improved YOLO face detection models. The mAP compares the ground-truth bounding box to the detected box and returns a score. The higher the score, the more accurate the model is in its detections. In a previous article, we looked in detail at the confusion matrix, model accuracy, precision, and recall. 

Average precision (AP) is a way to summarize the precision–recall curve into a single value representing the average of all precisions. The AP is calculated according to the next equation. Using a loop that goes through all precisions/recalls, the difference between the current and next recall is calculated and then multiplied by the current precision. In other words, the AP is the weighted sum of the precision at each threshold where the weight is the increase in the recall.

The model predicts the bounding boxes of the detected objects. It is expected that the predicted box will not match exactly the ground-truth box. The figure shows a face image. The ground-truth box of the object is in red while the predicted one is in yellow. Based on the visualizations of the two boxes, the model made a good prediction with a high match score. Figure 5 is regarding the InceptionResNet-v2.

Two transfer learning models were developed; the first model is based on the classification that predicts the head pose and the second model estimates the eye gaze direction.

Below Figure 6a,b are regarding the proposed head pose CNN detector and eye gaze estimation using classification and the CNN regression model. The data generations and extractions has been carried out by using the two modules which are divided into Figure 6a,b. The data analysis starts from both modules and proceeds accordingly. By the processing time the header and footer are added to the dataset as it goes towards the refining section. Three the data extractions take place and place it with the efficient gaze estimation method. From there the deep learning models are applied.

To fit the complete model into memory, older Inception models were trained in a partitioned method, with each copy split into numerous sub-networks. The Inception design, on the other hand, is highly adjustable, which implies that the number of filters in various layers may vary without affecting the quality of the completely trained network. It is used to carefully modify the layer size to balance the computations across several model sub-networks, to improve the training speed. The researchers were able to train the current model without splitting copies with the introduction of MATLAB. This was made possible in part by recent memory optimizations for backpropagation, which included carefully examining the tensor necessary for gradient calculation and designing the computation in such a way that the number of such feature vectors was reduced. 

The eye directions were estimated using the regression, a supervised learning approach with predictors in numerical type. The overall architectures of the CNN regression and CNN classification responsible for head pose detection were the same. The only differences were in the last layers. A regression model had two nodes in the output layer that estimated and predicted the eye gaze (vertical and horizontal angles). The feature learning block of the proposed CNN model was composed of 822 layers, while the last two layers were replaced with a regression layer. The model accepts data with a resolution of 299 × 299 and three-channel colors (RGB). The ADAM optimizer is an advanced minimization technique that selects the most feasible learning rate for the extracted CNN features. The optimal learning rate is used to compute the best weight for training data—the best weight calculation in an inefficient and time-consuming process that the proposed CNN model practices to achieve better eye gaze direction estimation performance. 

Researchers have always been cautious about modifying architectural decisions, restricting our experimentations to isolated network components while keeping the rest of the network stable. Researchers do not simplify their earlier decisions; they will end up with networks that are more complicated than they need to be. Researchers decided to dispense with this needless baggage in the latest trials for Inception-v4, creating a standard option for each grid-sized Inception block. The detailed structures of its components are shown in Figure 7 and Figure 8 while the large-scale topology of the Inception-v4 network is shown in Figure 9, All of the convolutions in the picture that are not indicated with a “V” are equally populated, meaning their output lattices are the same sizes as their input lattices. The convolutions indicated with a “V” were successfully filled, which indicates that each cell’s input patch is entirely included in the preceding layer, resulting in a lower grid size in the output activation map. Concerning blocks of residual Inception—researchers employed Inception blocks that were cheaper than the original Inception for the residual version of the Inception network. Before adding the input, each Inception block was followed by a filter expansion layer (1 convolution, no activation), which was used to increase the dimension of the filter bank to fit the depth of the input. This was required to compensate for the dimensionality reduction of the Inception blocks. Researchers have experimented with several Inception residual versions. Only two of them are discussed in depth in this article. 

The first, “Inception-ResNet-v1,” is approximately identical to the Inception-computational v3 cost, while “Inception-ResNet-v2” is similar to the recently announced Inception-v4 network. However, because of the larger number of layers, the Inception-step v4 size was proven to be substantially slower in reality. Another minor technical difference between our residual and non-residual Inception variations is that we employed batch normalization only on top of the standard layers, not on top of summation, in the case of Inception-ResNet. Although it is fair to believe that extensive batch normalization would be useful, we wanted each model copy to be trained on a single GPU. Layers with high activation sizes used a disproportionate amount of GPU RAM due to their memory footprints. We were able to greatly increase the overall number of Inception blocks by eliminating batch normalization on top of these layers. We expect that when computing resources are better exploited, this tradeoff will become obsolete. Figure 7 and Figure 8 are regarding the structure of the InceptionResNet-v2 network.

This is the Inception-A block of Figure 9.

### 3.3. Classification Model Performance Evaluation Metrics 

When evaluating the performance of our proposed model design, we used accuracy, precision, recall, and f-measure metrics to determine how well it worked. We decided to use the holdout data resampling method for the classification validation, with a random percentage of 70/30 used in the process. In this case, the accuracy of the classifier indicated the percentage of correct classifications; accuracy, in this case, was the inverse of the percentage of incorrect classifications. Regarding the positive observations, precision is defined as the proportion of correct positive observations over all positive observations. The percentage of times that a false positive was detected is referred to as recall, which is another term for sensitivity. It refers to the proportion of the expected positive events that actually occur in the manner anticipated. Precision and recall are combined to form the F-measure, which is a weighted average of the precision and recall; it represents the precision and recall combined. Therefore, both false positives and false negatives were taken into consideration when calculating this measure.
(1)$$\mathrm{Accuracy}=\frac{\left(\mathrm{TP}\;+\;\mathrm{TN}\right)}{\left(\mathrm{TP}\;+\;\mathrm{FP}+\;\mathrm{FN}+\mathrm{TN}\right)}$$
(2)$$\mathrm{Precision}=\frac{\mathrm{TP}}{\left(\mathrm{TP}\;+\;\mathrm{FP}\right)}$$
(3)$$\mathrm{Recall}=\frac{\mathrm{TP}}{\left(\mathrm{TP}\;+\;\mathrm{FN}\right)}$$
(4)$$F\text{-}\mathrm{Measure}=2*\frac{\left(\mathrm{Recall}\;*\;\mathrm{Precision}\right)}{\left(\mathrm{Recall}\;+\;\mathrm{Precision}\right)}$$

The performance evaluation metrics are given below for the proposed work.

Mean absolute error (MSE) is a measure of errors between paired observations expressing the same phenomenon.Root-mean-square error (RMSE) is the square root of the mean of the square of all of the errors. The use of RMSE is very common; it is considered an excellent general-purpose error metric for numerical predictions.The relative absolute error (RAE) is expressed as a ratio, comparing a mean error (residual) to errors produced by a trivial or naive model. A reasonable model (one that produces results that are better than a trivial model) will result in a ratio of less than one.

## 4. Experiments

### 4.1. Face Detection Results 

Figure 9 and Table 3 compare the results of the proposed modified YOLO-V4 network with the performance metrics of its corresponding network implementation. We coupled the previous work with the custom mask detection dataset stated above and split the dataset into 80% for the training set, 10% for the testing set, and 10% for the validation set, and trained and tested the network to obtain the performance metric values, as shown in Table 4. We trained and tested R-CNN, Fast R-CNN, Faster R-CNN, single shot detector (SSD), and improved YOLO-V4 on the employed mask detection dataset; we analyzed the performance metrics to compare the proposed network to its YOLO counterpart. When the performance metrics of the proposed modified YOLO-V4 were compared to those of other object detection networks used for the face detection task, the proposed method achieved good results with a mean accuracy (mAP) value of 64.31 %, which is 6.6% better than the original tiny YOLO-V4 and exceeds the performance of R-CNN, fast R-CNN, faster R-CNN, and SSD; the enhanced YOLO-V4 network described in this study is an incremental enhancement to the feature extraction network and overall performance of the original little YOLO-V4, displaying remarkable findings by recognizing the presence of masks in photos when masks are present everywhere on the facial area. 

The proposed network achieves an 84.42% average precision (AP) for the mask-like region, which is 14.05% higher than YOLO-V4 and higher than R-CNN, fast R-CNN, and faster R-CNN; thus, it enables the proposed network to detect almost all of the dataset’s small objects in images, such as masks, under different conditions. Furthermore, using the chosen dataset, the proposed upgraded YOLO-V4 network obtains 83.22% of the joint intersection (IoU) value, showing that the target detection network has excellent detection accuracy. In the intensely congested setting, the missing results also give a chance to investigate viable remedies. By calculating tiny anchor frames and training the network with picture samples from the occluded face, the detection accuracy and precision of such cases may be further enhanced. Figure 10 is regarding the performance comparison of different models.

### 4.2. Gaze Detection Results 

To test the suggested gaze analysis approach, a specially-built driven gaze picture dataset with seven classes was integrated into this part. The goal of our suggested technique was to create an image-based human gaze identification technique that identifies and forecasts the human eye’s head attitude and angle. A holdout cross-validation approach was used to split the dataset into a training set and a test set for performance assessment. Figure 11 and Figure 12 are regarding the training and validation of the gaze technique. While Figure 13 is regarding the confusion matrix.

The proposed model’s performance was assessed using the classification evaluation metrics of accuracy, precision, recall, and F-Measure in Table 5. For gaze detection, seven classes indicate head posture and eye orientation. A pre-trained CNN model for migration learning achieves higher accuracy, and CNN training takes one week owing to the size of the seven classes. For training the CNN, the epoch number is 50, the batch size is 64, and the optimization approach is Adam. The suggested model’s average accuracy is 92.71% and its average F-measure is 93.01%.

### 4.3. Eye Gaze Estimation Using CNN Regression

The class labels of eye direction are vertical and horizontal angles that are numerical values; these values can be predicted using a regression model. Our gaze consists of nine gaze directions, where each angle consists of different numbers of images. The proposed eye gaze estimation model was developed by performing transfer learning InceptionResNet-v2 and replacing the final two layers with a regression output layer. The hyperparameter used for model training was the optimization algorithm ‘Adam’, the number of epochs = 50, mini-batch size = 16, and an initial learning rate of 0.0001.

Figure 14, Figure 15 and Figure 16 are regarding the training and validation of RMSE and gaze estimation model.

To estimate the eye gaze angles, a regression model was utilized that predicted numerical values; the regression model could be linear or non-linear. As the data were linear, the proposed CNN model drew the line between the data points; the line passed between the data points, meaning that the model is the best fit and it receives a minimum RMSE score as illustrated in Table 6. 

Analysis of the proposed vertical eye gaze estimation model using the line plot.

Figure 17 depicts the results of the difference between the actual and predicted model’s vertical gaze angles using a line plot. The proposed model achieved a low RMSE of 2.68, which can also be visualized using the difference between the green and red markers in the line plot.

Figure 18 and Table 7 depicts the results of the difference between the actual and predicted model’s horizontal gaze angles using a line plot. The proposed model achieved a low RMSE of 3.61, which could also be visualized using the difference between the green and red markers in the line plot.

In the research article, a novel method for the driver gaze detection technique was developed using computer vision and deep learning-based technologies. The first task in this research involved developing a benchmark dataset; the dataset was composed of both labels, i.e., head pose detection and eye gaze estimation. There were seven classes of drivers’ head poses in the custom dataset, which were nominal labels, while the eye gaze estimation labels were vertical and horizontal angles (numeric values). The head pose detection in this research was performed using classification in machine learning while the eye gaze estimation was done using deep learning-based regression models. To accurately detect the gaze, it is important to select the face region only, which was performed using YOLO-V4 architecture. 

The existing architecture of YOLO-V4 was modified by replacing the CNN model, which is used for feature learning. As a base, the YOLO-V4 model Darknet-53 was used for feature learning. In this research work, the existing YOLO-V4 Darknet-53 architecture was replaced with Inception-v3 to extract more robust facial features and improve the performance of the drive face detection technique. For the head pose detection and eye gaze estimation, two parallel CNN models were used; the first CNN model detecting the head pose was a classification model while the second CNN model was a regression model that estimated the gaze directions vertically and horizontally. For the sake of developing these two models, various state-of-the-art transfer learning approaches were considered and validated by custom datasets. InceptionResNet-v2 achieved the highest accuracy for head pose detection and gaze estimation. For hyperparameter tuning, the model was trained several times with different hyperparameters. The developed model was tested in various challenging environments, i.e., low illumination, high illumination, noisy images, scale and translation changes of face regions; the proposed model performed the best by outperforming the CNN model by achieving the highest accuracy for head pose detection and the least RMSE for eye gaze estimation. 

The objective of this study was to create a research framework that could improve the performance of driver gaze detection using head poses and eye gazes; the first and most important step was to accurately detect the driver’s face, which was done using a YOLO object detector. For the driver, head pose transfer learning in the InceptionResNet-v3 CNN model was performed by replacing the number of nodes from 1000 to 7 and discarding all ImageNet data. Similarly, the same CNN model was used for eye gaze direction estimation; the difference between the CNN classification and the regression model is based on the last layer; the last two layers of the CNN classification model were replaced with a single regression layer of two nodes, where the first node estimated vertical eye angles and the second node was responsible for horizontal eye gaze angle estimation. The development of a custom benchmark dataset was the main objective of this study. A total of three datasets were developed; the first was used to train and validate the face detection system based on YOLO, the second dataset has seven classes, used for the head pose detection; the third (and final dataset) consists of face images with the driver’s eyes (horizontal and vertical directions).

## 5. Conclusions

Car crashes could lead to tremendous human and economic costs. Research shows that drivers are less likely (30–43%) to provoke collision-related damage when they have one or more passengers who can alert them. These reasons make it relevant to work on the development of an ADAT. The purpose of an ADAT is to support the driver in the driving process, improving car safety, particularly, and road safety, generally. The proposed technique consists of three deep learning models: the first model is an improvement of the YOLO feature learning block, which is used to detect the driver’s face in a high FPS video and segment it. Once the face region of the driver is segmented, it is passed to the following transfer learning models; the CNN classification model classifies the image and predicts a head pose out of seven driver head poses. The second model has the same architecture and feature learning block; the main difference is the replacement of CNN’s last two layers (with a regression layer of two nodes). The regression layer estimates the angles of the eye directions horizontally and vertically. All models used in the proposed framework achieved higher accuracy and low RMSE, which reflect their effectiveness and strength. The proposed system can be easily implemented and deployed in a car to monitor the various states and conditions of the driver, which could help reduce car accidents. 

## 6. Limitations of the Study

The proposed study is only limited to the gaze of a driver’s head. It does not calculate or extract the features from the whole body. Only deep learning CNN regression was used as the core contribution for this paper. The performance evaluation metrics were also limited. The study was limited in observing the pedestrian and driver’s eye contact in the real-time scenario. The study did not include or investigate all passengers in the vehicle, it only focused on the driver. The study also limited focus on the driver and the road, but not the whole vehicle, from the front/rear and the internal overall settings.

## Figures and Tables

**Figure 1 sensors-22-03959-f001:**
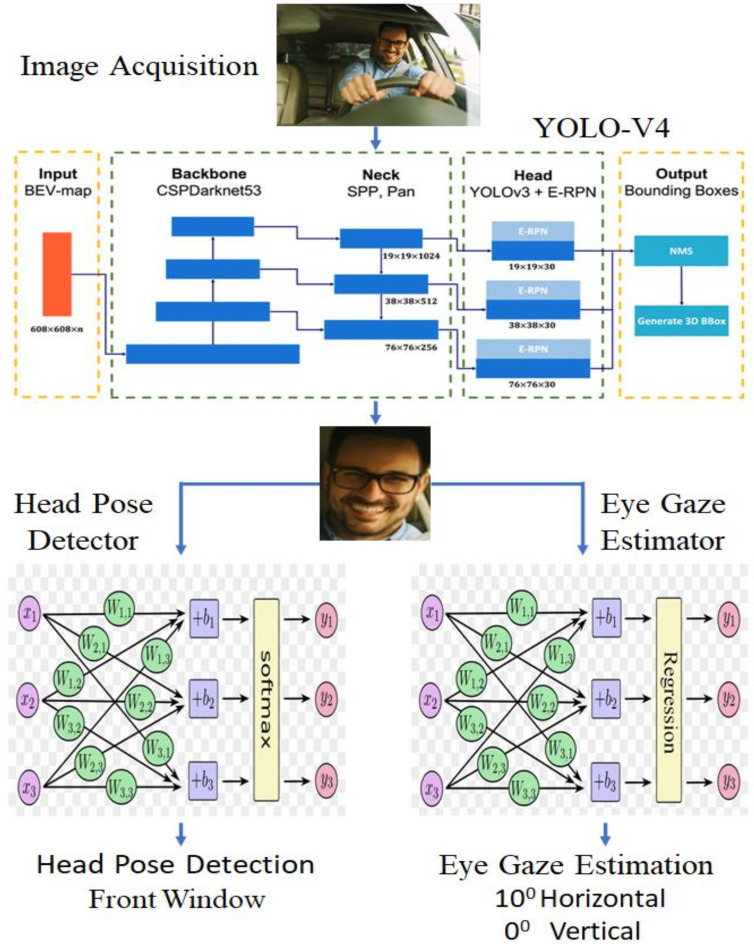
Proposed framework for driver gaze detection.

**Figure 2 sensors-22-03959-f002:**
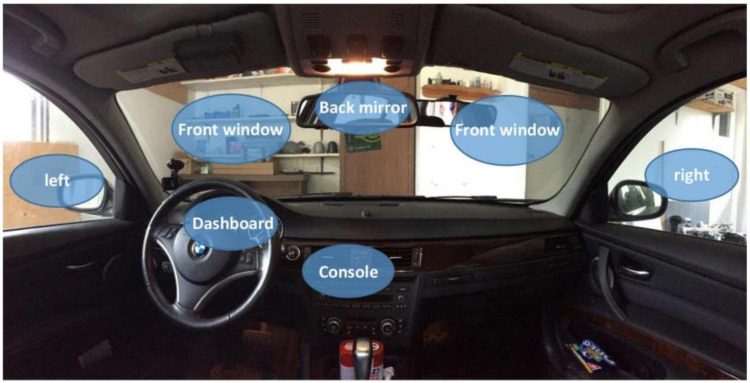
Selected face directional point of our dataset.

**Figure 3 sensors-22-03959-f003:**
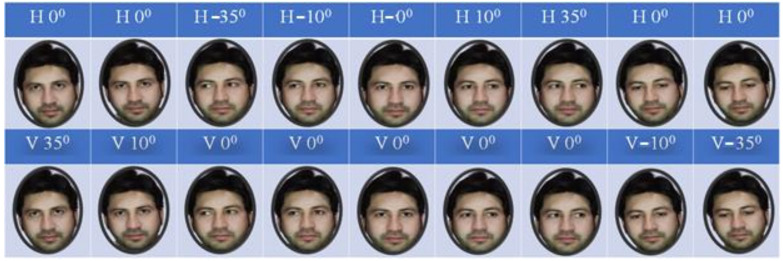
Selected eye gaze directional points of the custom-developed dataset.

**Figure 4 sensors-22-03959-f004:**
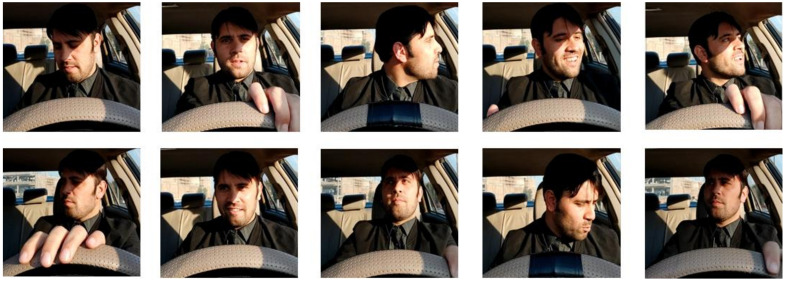
Random gazes recorded for the developing benchmark dataset.

**Figure 5 sensors-22-03959-f005:**
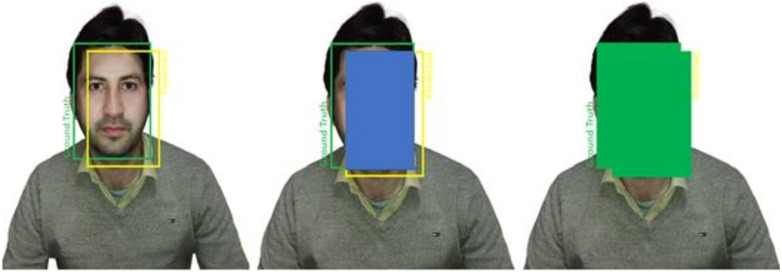
InceptionResNet-v2-based head and eye gaze estimation.

**Figure 6 sensors-22-03959-f006:**
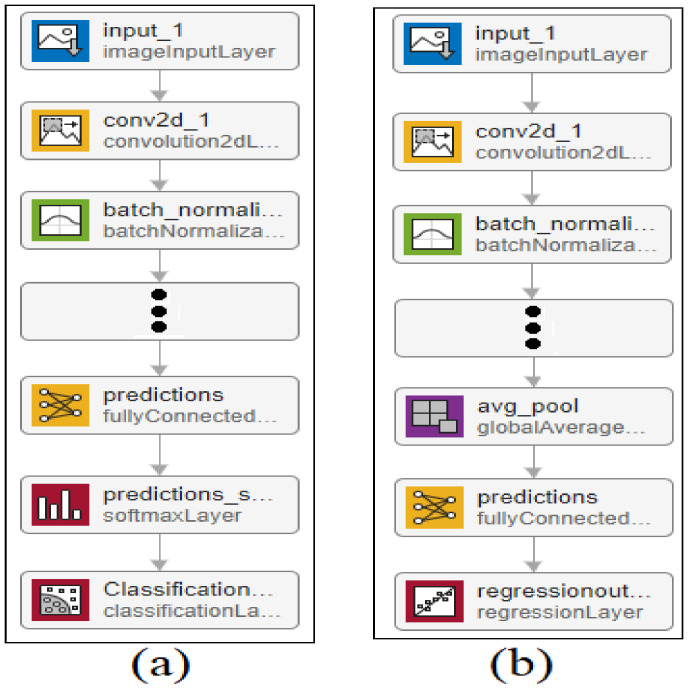
Proposed head pose CNN detector and eye gaze estimation using classification and the CNN regression model.

**Figure 7 sensors-22-03959-f007:**
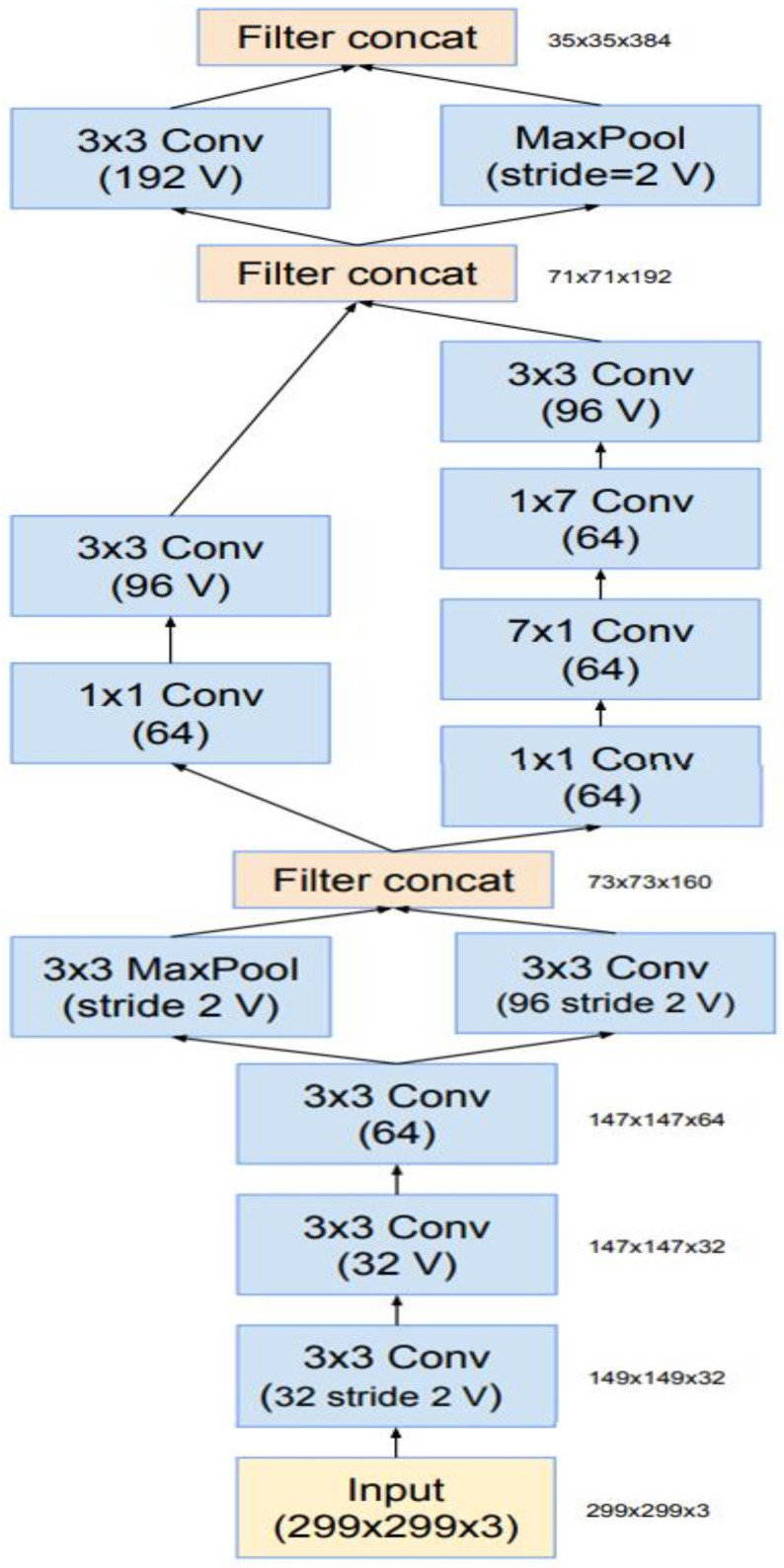
The schema for the stem of the pure Inception-v4 and Inception-ResNet-v2 networks. These are the input parts of those networks.

**Figure 8 sensors-22-03959-f008:**
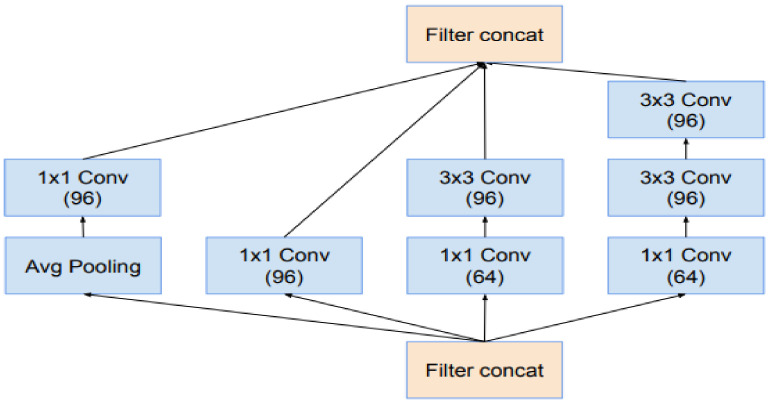
The schema for the 35 × 35 grid modules of the pure Inception-v4 network.

**Figure 9 sensors-22-03959-f009:**
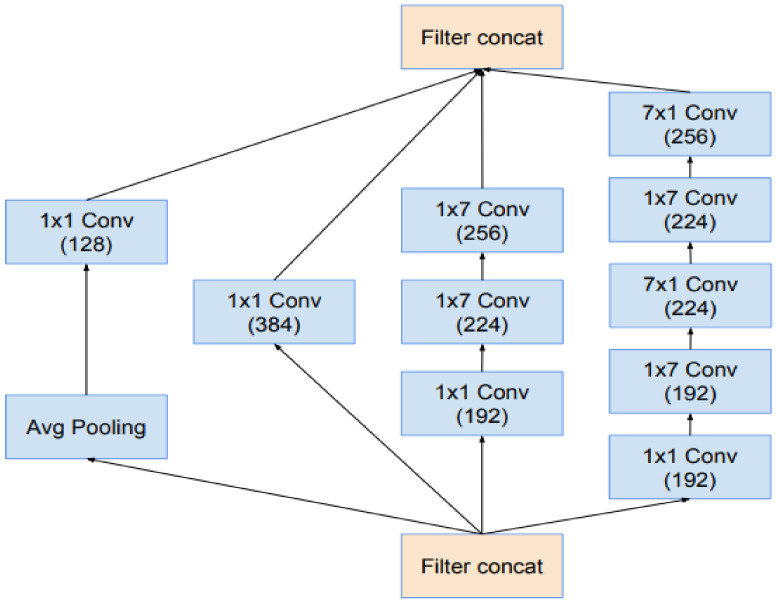
The schema for the 17 × 17 grid modules of the pure Inception-v4 network.

**Figure 10 sensors-22-03959-f010:**
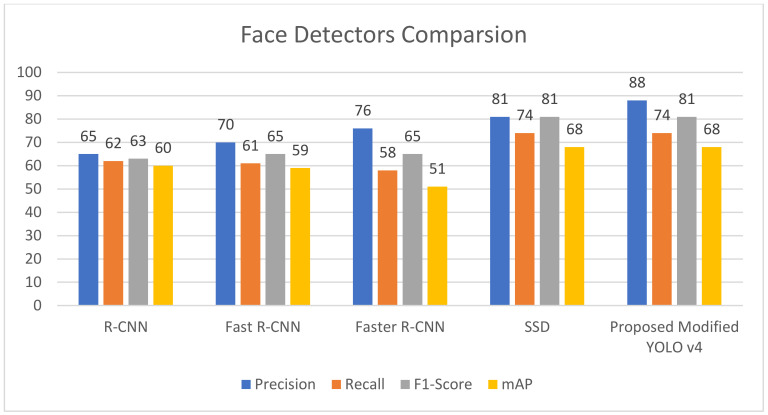
Performance comparison of various object detection techniques for face detection.

**Figure 11 sensors-22-03959-f011:**
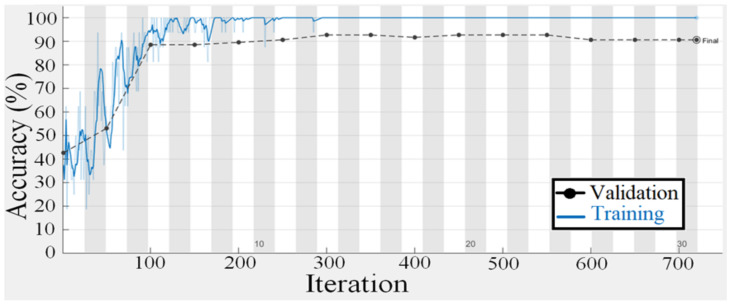
Training and validation accuracy plot of the gaze detection technique.

**Figure 12 sensors-22-03959-f012:**
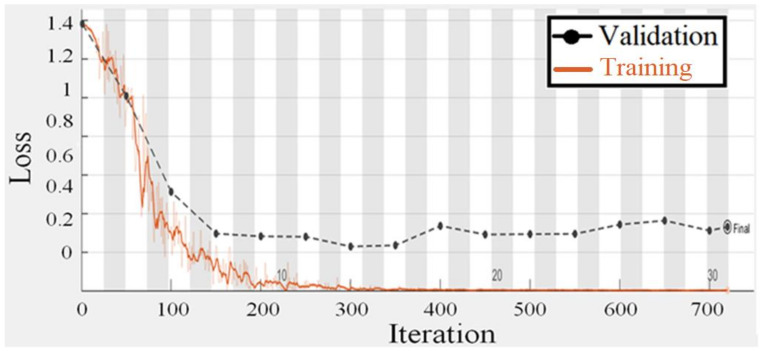
Training and validation loss plot of the gaze detection technique.

**Figure 13 sensors-22-03959-f013:**
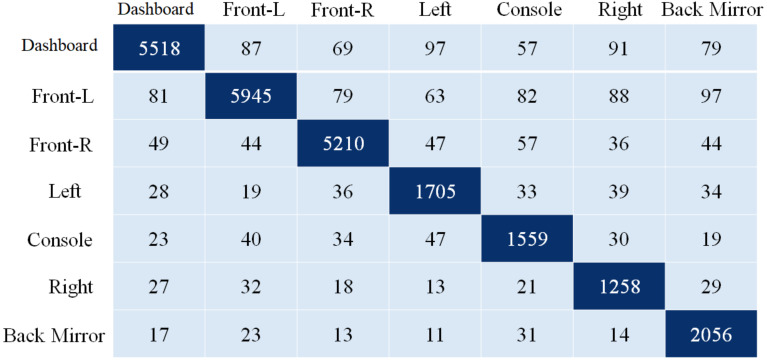
Confusion matrix of the proposed driver gaze detection technique.

**Figure 14 sensors-22-03959-f014:**
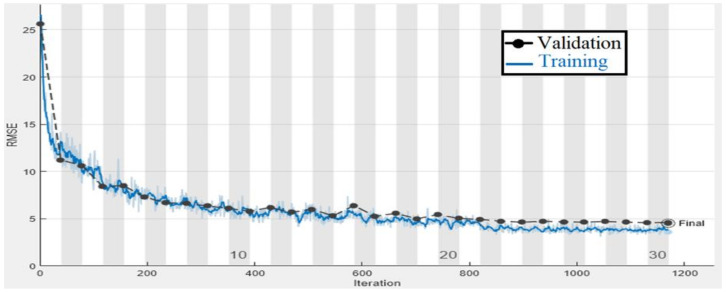
Training and validation RMSE plot of the eye gaze direction estimation model.

**Figure 15 sensors-22-03959-f015:**
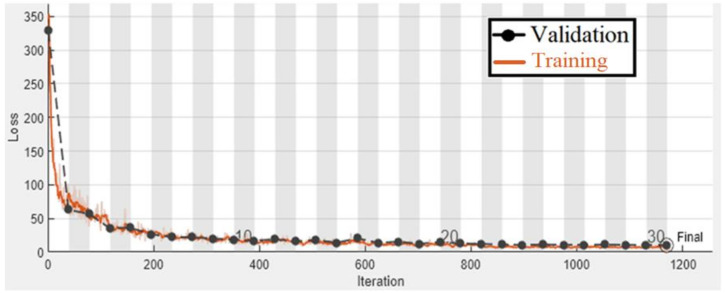
Training and validation loss plot of the eye gaze direction estimation model.

**Figure 16 sensors-22-03959-f016:**
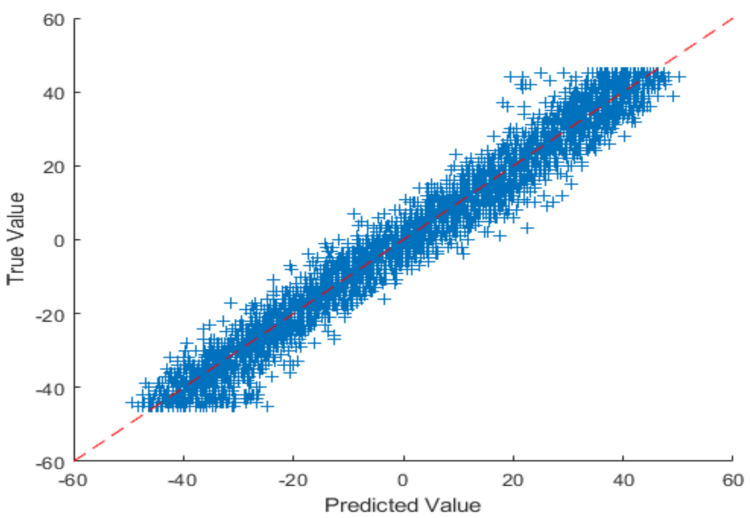
Eye gaze estimation model curve.

**Figure 17 sensors-22-03959-f017:**
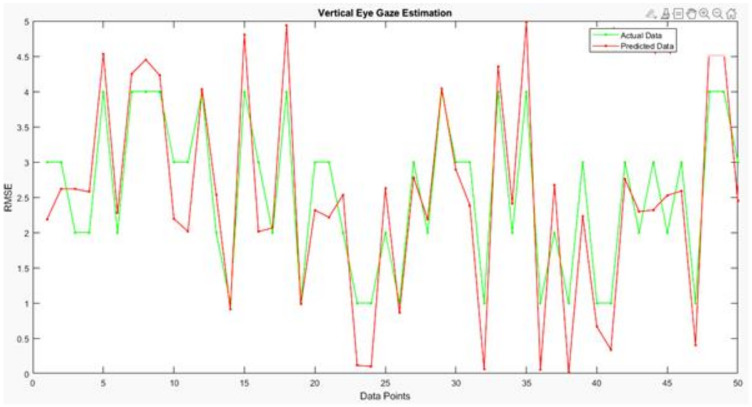
Analysis of the proposed vertical eye gaze estimation model using the line plot.

**Figure 18 sensors-22-03959-f018:**
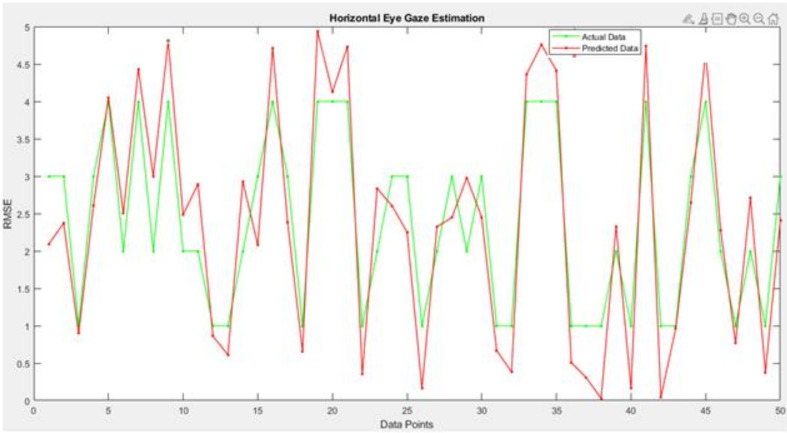
Analysis of the proposed horizontal eye gaze estimation model using the line plot.

**Table 1 sensors-22-03959-t001:** Custom gaze detection dataset details.

Class	Training (70%)	Testing (30%)
Dashboard	13,995	5998
Front-L	15,015	6435
Front-R	12,803	5487
Left	4419	1894
Console	4018	1722
Right	3262	1398
Back-Mirror	5051	2165

**Table 2 sensors-22-03959-t002:** Custom eye gaze estimation dataset details.

Class	Training (70%)	Testing (30%)
H 0°–V 35°	10,063	4313
H 0°–V 10°	11,554	4952
H −35°–V 0°	10,938	4688
H −10°–V 0°	8834	3786
H 0°–V 35°	9195	3941
H 0°–V 0°	10,502	4501
H 10°–V 0°	10,213	4377
H 35°–V 0°	7606	3260
H 0°–V −10°	7730	3313
H 0°–V −35°	8150	3493

**Table 3 sensors-22-03959-t003:** Performance comparison of object detection techniques on the CUSTOM dataset.

Various Techniques	Precision(%)	Recall(%)	F-1 Score(%)	mAP(%)
R-CNN	65	62	63	60
Fast R-CNN	70	61	65	59
Faster CNN	76	58	65	51
SSD	81	67	74	59
Proposed YOLO-V4	85	74	81	68

**Table 4 sensors-22-03959-t004:** Performance comparison between YOLO-V4 Darknet-53 and YOLO-V4 Inception-v3.

Model	Appearance	FPS	mAP
YOLO-V4 + Darknet-53	No Glasses & No Beard	59	63
YOLO-V4 + Darknet-53	Beard	54	57
YOLO-V4 + Darknet-53	Glasses	51	55
YOLO-V4 + Darknet-53	Glasses & Beard	46	52
YOLO-V4 + Inception-v3	No Glasses & No Beard	66	68
YOLO-V4 + Inception-v3	Beard	60	59
YOLO-V4 + Inception-v3	Glasses	58	56
YOLO-V4 + Inception-v3	Glasses & Beard	54	55

**Table 5 sensors-22-03959-t005:** Detail percentage by class of various driver gazes.

Class	Accuracy	Precision	Recall	F-Measure
Dashboard	98.5	87.5	91.5	91.5
Front Window-L	93.28	86.2	93.3	91.7
Front Window-R	98.64	93	98.6	97.8
Left	96.85	85.2	96.9	92.9
Console	95.07	89.5	95.1	94.3
Right	91.5	81.7	91.5	88.5
Back Mirror	86.14	90.2	86.1	90
Average	92.71%	87.61%	93.29%	93.01%

**Table 6 sensors-22-03959-t006:** Performance of CNN regression for the vertical eye gaze.

Evaluation Matrix	Result in % Age
Correlation coefficient	0.9459%
Mean absolute error	0.5748%
Root mean square error	2.6853%
Relative absolute error	8.6104%
Root relative square error	32.8844%

**Table 7 sensors-22-03959-t007:** Performance of CNN regression for the horizontal eye gaze.

Evaluation Matrix	Result in % Age
Correlation coefficient	0.9347%
Mean absolute error	0.9423%
Root mean square error	3.6127%
Relative absolute error	10.9899%
Root relative square error	36.1281%

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
