# Peer review of "A Driver Gaze Estimation Method Based on Deep Learning"

_sensors, 2022, doi:10.3390/s22103959_

Round 1
Reviewer 1 Report
In the present manuscript, the authors systematically describe how they develop a deep learning method that notifies the driver under a dangerous scenario, the Advanced Driver Assistance Technique (ADAT) which is based on gaze tracking and estimation. The authors describe all the steps in detail, from the role of head poses and eye gaze directions in different circumstances until the procedure they adopted to detect the driver’s face with sufficient accuracy. The latter is performed by the named YOLOv4 face detector which is described as an improved convolutional neural network (CNN) inceptionv3. Other details are also discussed such as how they employed transfer learning from another version of CNN inceptionresnetv2.
I think that the work is important, especially due to the goal of avoiding car accidents. Having said that, there is a number of issues I see in the current manuscript that I think the authors should address to improve their work and the clarity of the presentation. Please read below:
Major comments:
- Introduction, second paragraph: many different works are discussed here but the paragraph is poorly cited. Whenever the authors bring a discussion from another work they should include the proper citation. This happens in other places in the paper too, as for example in the literature review. Please revise.
- Lines 197-210: there are many phrases that deserve a revision. Some phrases need to be restructured as they are hard to understand. Some minor errors are due to grammar, e.g.: “A numerous researches” -> “Numerous researchers (and add citation), “The authors of [26] used with “, “and found in which the pedal response of a driver”. Please revise.
- Please define in the methods how mAP (mean accuracy) is defined. Also, other averaged metrics have to be defined.
- It is not entirely clear how the authors use the training, validation, and test sets. Also, it is not entirely clear from which of the sets these metrics are taken. In general, it’s good practice to take metrics such as accuracy, f1-scores, precision, and recall from the validation and not from the training set. If the authors are taking from the training this has to be explained.
- Figs 10, 11, 13, and 14 have a similar problem. There are two curves and the caption says “training and validation”, but there is no mention of which curve is the training and validation. Similarly, why would the authors take measures from the training set when they should be taken from the validation. And how do the authors use the testing set and where?
- The discussion on Figs 15-17 is nearly absent when nonexistent. The Authors should properly address a discussion around these figures.
- Line 551 – “which will be validated in real environment”-> wouldn’t “tested” be a more appropriate word.
Minor comments:
- Introduction: Please introduce ADAT here too.
- Introduction, line 45: please check “[29-41”. Is this a citation or a percentage?
- Line 256: “bearded or unbearded” is repeated twice. Please remove one mention.
- Line 321: “Figure 2 depicts”, do you mean Figure 1?
- Line 364: “The researchers were able to train current model” -> “to train the current model”.
- Lines 368-370: please revise the statement and correct grammar. Same for lines 379-381.
- Line 371: “to do rid” -> “to get rid”.
- Line 388: “because to the larger” -> “because of the larger”.
Author Response
Dear Reviewer
Thank you for giving us the opportunity to submit a revised draft of our manuscript titled “A Driver Gaze Estimation Method Based on Deep Learning” to Sensors Journal. We appreciate the time and effort that you have dedicated for providing your valuable feedback on our manuscript.

Reviewer 2 Report
The authors proposed a new solution for recognizing people's head and gaze positions while driving a car. They adopted the method known from the literature and conducted tests comparing their solution with other approaches, receiving better results. However, the paper requires improvement in terms:
- description of the dataset used; the number of samples, their shape etc.,
- the description of the labeling samples process, - an explanation why the professional photographer took head photos? Can it influence the performance when real, not-so-perfect data are used?
- the time performance - showing the feasibility of the method's usage in real scenarios
- the presentation on how to utilize the obtained results,
- the paper language.
Additionally, descriptions and labels in tables and charts have to be improved.
Author Response

(The authors gave the same response as above.)

Round 2
Reviewer 1 Report
I appreciate that the authors have addressed all my questions and improved the content of their work. I believe the manuscript can be accepted.
Author Response
Dear Reviewer
Thank you for giving us the opportunity to submit a revised draft of our manuscript titled “A Driver Gaze Estimation Method Based on Deep Learning” to Sensors Journal. We appreciate the time and effort that you have dedicated for providing your valuable feedback on our manuscript. We are grateful to you for your insightful comments on our paper. We have been able to incorporate changes to reflect most of the suggestions provided by you. We have highlighted the changes within the manuscript.
Here is a point-by-point response to the editor comments and concerns.
Comments from Reviewer:
- Comment 1: I appreciate that the authors have addressed all my questions and improved the content of their work. I believe the manuscript can be accepted.
Response: We are thankful to your review and please that you recommended acceptance on our manuscript.
Sincerely,
Corresponding Author

Reviewer 2 Report
Dear Authors,
Thank you for your response. However, I cannot find extended information about the dataset in the uploaded text - how many samples were in the datasets, and how many elements constitute feature vectors? The same regards the possible applications of your solution. How do you think it could be included in the current car's equipment.
Please mark it in yellow in the manuscript.
Author Response
Dear Reviewer
Thank you for giving us the opportunity to submit a revised draft of our manuscript titled “A Driver Gaze Estimation Method Based on Deep Learning” to Sensors Journal. We appreciate the time and effort that you have dedicated for providing your valuable feedback on our manuscript. We are grateful to you for your insightful comments on our paper. We have been able to incorporate changes to reflect most of the suggestions provided by you. We have highlighted the changes within the manuscript.
Here is a point-by-point response to the editor comments and concerns.
Comments from Reviewer:
- Comment 1: However, I cannot find extended information about the dataset in the uploaded text - how many samples were in the datasets, and how many elements constitute feature vectors? The same regards the possible applications of your solution. How do you think it could be included in the current car's equipment.
Please mark it in yellow in the manuscript.
Response: We are thankful to your review, the both custom developed datasets description is added in the updated manuscript. The application of driver gaze detection system and deployment procedure are also explained in details. All the recommended changes are heighted.
We look forward to hearing from you in due time regarding our submission and to respond to any further questions and comments you may have.
Sincerely,
Corresponding Author
